# OpenReview forum: "Decentralized Consent Orchestration: A Framework for Auditable, Revocable, and Forward-Compatible Data Sharing in Solid"
_SolidProject.org/SoSy/2025/Privacy_Session — Submitted to SoSy2025-Privacy_

### Official Review · ~Konrad_Kollnig1 · 2025-03-19
**Compelling idea that needs more thorough argumentation and analysis**

**Rating:** 6
**Confidence:** 4

**Review:**

The paper presents a compelling idea but falls short in making an argument in support of their approach. A thorough analysis of the problem, supported by evidence, seems absent. This, however, would be important to define the correct requirements ahead of developing a consent system. Despite these shortcomings, I recommend acceptance because I think it'd provide a good foundation for discussion.

The introduction should make very clear why this is needed but I'm not convinced. Why do we need this consent solution? Why does it have to be decetralised? How does it compare to existing consent approaches like TCF (which is likely incompatible with GDPR, as found by the CJEU)?

The paper makes claims about consent management platforms but this analysis remains superficial. Also, a discussion of the industry standard in Europe, TCF, is absent.

Since this paper tries to consider a legal dimension, I'd appreciate a thorough engagement with how the current legal framework falls short in practice, and why this needs a technical solution. Isn't there already enough compliance to go through for companies? The EU seems to agree and is considering reducing the complexity of the GDPR. Therefore, I'm not fully convinced that consent is the main problem that we face right now, but I'm happy to be convinced otherwise. However, this would require thorough engagement with the status quo in law and compliance, and the identification of suitable requirments for the software system.

"Evaluation through a prototype implementation and user studies shows that DCO enables more compliant". How exactly did this evaluation of compliance involve legal experts? You claim that you evaluated something in Section 5 but do not provide much detail.

The writing of this paper is confusing to me and contains quite a few bullet points. There also could be more references. For example, the statement "Legal frameworks require consent to be specific, informed, unambiguous, and revocable" seems to relate to the GDPR but this isn't mentioned.

---

### Decision · Program_Chairs · 2025-04-01

Reject